# The Role of Hypoxic Bone Marrow Microenvironment in Acute Myeloid Leukemia and Future Therapeutic Opportunities

**DOI:** 10.3390/ijms22136857

**Published:** 2021-06-25

**Authors:** Samantha Bruno, Manuela Mancini, Sara De Santis, Cecilia Monaldi, Michele Cavo, Simona Soverini

**Affiliations:** 1Department of Experimental, Diagnostic and Specialty Medicine, University of Bologna, 40138 Bologna, Italy; samantha.bruno2@unibo.it (S.B.); sara.desantis9@unibo.it (S.D.S.); cecilia.monaldi2@unibo.it (C.M.); michele.cavo@unibo.it (M.C.); 2Istituto di Ematologia “Seràgnoli”, IRCCS Azienda Ospedaliero, Universitaria di Bologna, 40138 Bologna, Italy; manuela.mancini6@unibo.it

**Keywords:** acute myeloid leukemia, hypoxic bone marrow microenvironment, metabolic reprogramming, leukemic stem cell

## Abstract

Acute myeloid leukemia (AML) is a hematologic malignancy caused by a wide range of alterations responsible for a high grade of heterogeneity among patients. Several studies have demonstrated that the hypoxic bone marrow microenvironment (BMM) plays a crucial role in AML pathogenesis and therapy response. This review article summarizes the current literature regarding the effects of the dynamic crosstalk between leukemic stem cells (LSCs) and hypoxic BMM. The interaction between LSCs and hypoxic BMM regulates fundamental cell fate decisions, including survival, self-renewal, and proliferation capacity as a consequence of genetic, transcriptional, and metabolic adaptation of LSCs mediated by hypoxia-inducible factors (HIFs). HIF-1α and some of their targets have been associated with poor prognosis in AML. It has been demonstrated that the hypoxic BMM creates a protective niche that mediates resistance to therapy. Therefore, we also highlight how hypoxia hallmarks might be targeted in the future to hit the leukemic population to improve AML patient outcomes.

## 1. Introduction

Acute myeloid leukemia (AML) is a highly heterogeneous hematologic malignancy that results from a wide range of alterations of myeloid precursors responsible for the uncontrolled proliferation, block of differentiation, and escape from apoptosis. The resulting expansion of leukemic-initiating cells induces bone marrow (BM) failure, leading to normal hematopoietic functions. AML represents the most common acute type of leukemia in adults, affecting approximately 1% of the population [1]. Although AML can occur at any age, the disease incidence increases sharply with age, showing a median age at diagnosis of 68 years.

AML originates from the oncogenic transformation of hematopoietic stem cells (HSCs) that may undergo further genetic alterations leading to the progression of the disease, with multiple coexisting clones that evolve over time [2,3]. The disease phenotype and its clinical course are strikingly influenced by several recurrent mutations, which define the prognostic stratification of patients. Some recurrently mutated genes are now potential targets for a series of newly approved or investigational treatments [4,5,6]. Since 1970, the standard of care for AML patients has been based on chemotherapeutic treatment (the ‘3 + 7’ regimen, combining daunorubicin and cytarabine), resulting in 5-year survival rates of 40–45% for patients up to 55 years, that decrease to 30–35% for patients up to 60 and may become as low as 10–15% in older patients [7,8]. Moreover, the relapsed disease after complete remission still represents the most common cause of death among AML patients, and it is associated with increased molecular complexity that often contributes to treatment failure [9,10]. Therefore, unraveling the cytogenetic and molecular profile of AML has improved the therapeutic opportunities leading to the development of new targeted therapies.

In recent years, however, it has been acknowledged that AML development and progression are dependent on cell-intrinsic oncogenic alterations and are also strongly influenced by the components of the bone marrow microenvironment (BMM) where normal and leukemic stem cells (LSCs) reside. It has been demonstrated that, physiologically, the BMM can be distinguished in two distinct niches, defined endosteal and vascular niche [11,12,13]. The endosteal HSC niche is mainly composed of osteoblasts that support hematopoiesis [11,14], while the HSC vascular niche mainly contains sinusoidal endothelial cells [12], which support HSCs proliferation and differentiation [15,16]. It has also been demonstrated that low pO_2_ levels characterize the BMM, although highly vascularized [17,18,19]. In particular, it has been shown that O_2_ levels in BM range from 1% to 4% [17,20], decreasing from vessels to the endosteum—hence the term “hypoxic niche.” Tumor hypoxia has been first described as a pathogenic condition in most solid cancers. It has been associated with increased tumor aggressiveness, immunosuppression, and decreased sensitivity to radiotherapy and chemotherapy [21]. The earliest evidence about the presence of a hypoxic niche arose from both in vivo studies, which confirmed that HSCs reside in hypoxic BM areas with diminished blood perfusion [22,23,24,25], and from in vitro hypoxic cultures (1%–3% O_2_) that elucidated the role of hypoxia in promoting quiescence and self-renewal of HSCs [26,27,28], as well as their differentiation into erythroid, megakaryocytic, and granulocytic-monocytic progenitors [29,30,31,32]. The peculiar property of hypoxic BMM is that it represents a physiologic condition to control HSCs homeostasis. Still, in the case of leukemic transformation, it becomes inhospitable to normal HSCs creating a ”malignant niche” that sustains survival and proliferation of LSCs [33]. A growing number of studies have provided valuable insights into the crucial role of hypoxia in the pathophysiology of AML. The increasing interest in the hypoxic niche arises from its dual role in regulating fundamental cell fate decisions for HSCs and LSCs. Therefore, in this article, we will review the current understanding of the pathogenetic impact of hypoxic BMM in AML, and we will outline some hypoxia-targeted treatment strategies that could be further developed in the future.

## 2. Regulation of Response to Hypoxia

Hypoxia-inducible factor (HIFs) proteins mediate the major adaptive response to hypoxia by introducing many genes required for adaptation to hypoxia. The HIF family includes three O_2_-sensitive α-subunits (HIF-1α, HIF-2α, and HIF-3α) and one stable β-subunit (HIF-1β/ARNT). They work as a heterodimer through the interaction between α and β subunits. In normal O_2_ conditions, the two α subunits are hydroxylated by the interaction with prolyl hydroxylases (PHDs) located on von Hippel-Lindau-associated E3-ubiquitin ligase (VHL), and then are degraded by the proteasome to ensure a rapid turnover. On the contrary, under hypoxic conditions, the PHDs are inactive because they require molecular oxygen as a cofactor. Thus, the binding of α subunits to VHL is inhibited. The final consequence is stabilizing the oxygen-sensitive subunit and its translocation into the nucleus, where it can dimerize with HIF-1β. This heterodimer, in turn, interacts with the transcriptional co-activator p300/CREB-binding protein (CBP) to recognize the hypoxia-responsive elements (HREs), leading to the genetic expression designed to boost the adaptive response to hypoxia [34] (Figure 1). The expression of HIF downstream genes has been associated with many cellular functions, including angiogenesis, energetic metabolism (glycolysis, tricarboxylic acid cycle, and oxidative phosphorylation), proliferation and survival, autophagy, proteolysis, and pH regulation [35,36,37].

## 3. Effects of Hypoxia and HIFs on HSCs

Since the concept of the hypoxic microenvironment was proposed, many studies have investigated the role of hypoxia role on HSCs functions, building from previous observations on solid cancers that demonstrated the crucial impact of HIF proteins on cellular transcriptional programs. Several in vivo imaging studies have demonstrated that HSCs preferentially localize in poorly perfused endosteal niches [38,39] that promote quiescence and pluripotent state of HSCs [40] and support their expansion in response to damage signals [23,41,42]. In contrast, it was observed that HSCs residing in the vascular niche are short-term proliferating cells responsible for replenishing circulating cells [16,38] (Figure 2). Furthermore, an immune-protective property of hypoxic niches has been documented mediated by regulatory T cells (Treg) that preferentially localize in the endosteal niche where they shelter HSCs from CD4 and CD8 T cell immune attack, enabling transplanted allo-HSCs to escape from allogeneic rejection [43,44].

Among the three α subunits, HIF-1α and HIF-2α have been more comprehensively studied, while less is known about HIF-3α. HIF-1α expression in HSCs is transcriptionally controlled by the homeodomain transcription factor Meis1 [45], while HIF-2α is a target of Stat5 [46]. In accordance with the hypoxic environment of the BM, it has been reported that normal HSCs show enrichment of HIF-1α [45,47] and that its genetic deletion results in loss of quiescence and reduction of repopulating activity of HSCs [47,48]. Moreover, through in vivo and in vitro studies, it has been demonstrated that HIF-1α accumulation obtained by hypoxic culture conditions, as well as its genetic stabilization deriving from VHL deletion or functional inhibition of the PHD domain, is enough to induce HSC quiescence and to enhance their self-renewal potential [49,50,51]. Loss-of-function studies indicated that HSCs isolated from HIF-1α-deficient mice showed increased expression of the hallmark genes of senescent stem cells p16^Ink4a^ and p19^Arf^ [47]. Therefore, much evidence supports the crucial role of HIF-1α as a master regulator of quiescence, self-renewal capacity, and survival of HSCs [52]. As previously mentioned, most of the adaptive changes induced by HIF-1α have been associated with the expression of several genes involved in numerous cellular processes, making them important players for HSC function. They include those encoding for the vascular endothelial growth factor (*VEGF*), stromal cell-derived factor 1 (*SDF1*), stem cell factor (*SCF*), angiopoietin 2 (*ANGPT2*), angiopoietin-like proteins (*ANGPTLs*), insulin-like growth factor 2 (*IGF-2*), insulin-like growth factor-binding proteins (*IGFBPs*), forkhead box O (*FOXOs*) and the cyclin-dependent kinase inhibitors p16, p19 and p21 [35,36,37,47,53]. In addition, HIF-1α has been deeply characterized as a metabolic regulator, able to favor the switch from oxidative to glycolytic metabolism, thereby limiting the mitochondrial potential to allow the quiescence state of HSCs [54,55,56]. Indeed, HIF-1α activates the transcription of glucose transporters, glycolytic enzymes, and glycolytic inducing factors [54,57]. Its conditional deletion in HSCs results in a metabolic shift to oxidative metabolism, increasing oxidative stress [58].

While the crucial role of HIF-1α on HSC maintenance has been clarified, some incertitude remains regarding the exact role of HIF-2α. HIF-2α deletion does not change HSC number or steady-state hematopoiesis [59]. However, it induces increased reactive oxygen species (ROS) production, which in turn stimulates endoplasmic reticulum stress and apoptosis together with a considerable engraftment reduction during xenotransplantation [60].

Collectively, these studies demonstrated that the hypoxic BMM sustains the cellular homeostasis of HSCs through the regulation of the gene expression programs mostly mediated by HIFs proteins.

## 4. HIFs in AML

The peculiar property of hypoxic BMM is related to its ambivalent role in physiologic and pathologic conditions. Therefore, understanding the complex interplay between the hypoxic microenvironment and the different cellular populations that reside in the BM could contribute to identifying pathway alterations leading to leukemic transformation. Thus far, many studies have demonstrated the involvement of BMM in the initiation and propagation of leukemia. Several pieces of evidence concur to demonstrate that the hematopoietic niches sustain survival, proliferation, and differentiation of LSCs [12,13,14,61,62,63]. A series of studies have also shown that the crosstalk between LSCs and the BM niche plays a dual role, being involved in both leukemia-induced microenvironment reprogramming and microenvironment-induced leukemogenesis mechanisms. Indeed, it has reported that engraftment and proliferation of CD34 + LSCs in BM induce alterations in the stromal microenvironment, creating a “malignant niche” inhospitable to normal HSCs [64,65].

Regarding the expression of HIF-1α in AML, it has been demonstrated that out of 84 cases of normal karyotype AML, about 20% displayed leukemic cells expressing high levels of cytoplasmic HIF-1α [66]. In another study, Wang et al. investigated HIF-1α expression in AML, showing that HIF-1α is up-regulated in CD34 + CD38-human AML primary cells. Moreover, they demonstrated that *HIF-1α* silencing and its inhibition by echinomycin induced apoptosis in LSCs and impaired their ability to reconstitute AML into xenotransplanted mice [67]. As far as the studies that have explored the expression levels of HIF-2α are concerned, it has been reported that in a cohort of 33 AML cases, 10 displayed higher levels of HIF-2α compared to normal CD34 + cells [60]. Moreover, the same study showed that *HIF-2α* silencing in leukemic blasts significantly reduced the leukemic engraftment into immunodeficient mice [60]. Finally, Vukovic et al. observed that *HIF-2α* deletion accelerated leukemia initiation in mixed-lineage leukemia fusion driven (*MLL-AF9*) mice models and that this acceleration was further potentiated by *HIF-1α* co-deletion [68].

This evidence indicates that understanding the crosstalk between LSCs and BM niche is essential to better understand the leukemogenic process and some therapy resistance/evasion mechanisms adopted by LSCs. It provides the rationale to develop novel therapeutic approaches to target the cytoprotective mechanisms deriving from the “malignant niche.”

## 5. Metabolic Alterations Induced by Hypoxia

One of the best-described alterations induced by hypoxia is the regulation of energetic metabolism, which is susceptible to a marked change mostly mediated by the transcriptional regulation of several key enzymes. Hypoxia and HIF family proteins have already been demonstrated to affect cellular metabolism through the switch from mitochondrial oxidative phosphorylation (OXPHOS) to glycolysis to avoid oxidative stress related to OXPHOS function in low O_2_ levels conditions. Metabolic analyses performed in AML cell lines and primary blasts confirmed that the major metabolic changes associated with the hypoxic microenvironment include those involved in energy production. However, the available data in AML models are still controversial, suggesting there might be a certain degree of dependence on specific cell-intrinsic factors [69]. On the one hand, Lodi and colleagues [70] described similar metabolic effects induced by hypoxia in two different models of leukemia, the AML cell line KG-1 and the chronic myeloid leukemia (CML) cell line K562. They observed that, although the two cell lines have different metabolic profiles in normoxia, the metabolic changes were strikingly similar when cultured under hypoxia (1% O_2_), suggesting a common adaptive response to hypoxia. In particular, the study described that hypoxia induces enrichment of the glycolytic pathway with increased levels of lactate and alanine and citrate, fumarate, and succinate, which are tricarboxylic acid (TCA) metabolites [70]. Moreover, both cell lines showed increased concentration of metabolites of the hexosamine pathway such as uridine diphosphate *N*-acetylglucosamine (UDP-GlcNAc) and uridine diphosphate *N*-acetylgalactosamine (UDP-GalNAc) [70], which was previously associated with response to several cellular stresses, including hypoxia [71].

On the other hand, we compared the metabolomic alterations induced by hypoxia on OCI-AML3 and KASUMI-1 AML cell lines and showed that hypoxic adaption might rather be dependent on the genomic background [72]. After 20 h of hypoxia, OCI-AML3 cells showed reduced pyruvate levels. They increased 2-hydroxyglutarate (2-HG), fumarate, succinate, lactate, and alanine levels, whereas KASUMI-1 exhibited increased levels of alanine, glutamine, glutamate, and fatty acids with little impact on the TCA cycle intermediates [72]. Lastly, the study published by Goto et al. analyzed the differences in growth and survival of the acute monocytic leukemia THP-1 and the acute promyelocytic leukemia NB-4 cell lines under hypoxic culture conditions. They reported that whereas NB4 cells grew slowly and showed increased ROS production after 48 h of hypoxia, THP-1 cells quickly adapted to hypoxic culture conditions, avoiding ROS production. They did so by changing their energy metabolism from OXPHOS to glycolysis through the up-regulation of pyruvate dehydrogenase kinase 1 (PDK1), a key glycolytic enzyme that inhibits the conversion of pyruvate to acetyl-coenzyme A.

Moreover, the two cell lines also showed different adaptation mechanisms against ROS-induced damage. NB4 cells cultured for longer periods (7 days) increased the ratio between reduced and oxidized glutathione (GSH/GSSG) in order to avoid oxidative stress, while hypoxic THP-1 treated with a glycolysis inhibitor to force ROS production, showed only a slight reduction of growth without accumulation of ROS, given their ability to exchange the cytochrome C oxidase subunit 4 isoform 1 (COX 4-1) subunit to COX 4-2 [73]. Thus, these two studies conducted in the AML cell lines showed that the specific adaptive changes differed strikingly among the different AML cellular models, highlighting a potential role of the genomic background during adaption to hypoxia.

The metabolic reprogramming of AML cells and their heavy dependence on glucose consumption were confirmed in primary leukemic cells [69,74]. A study by Chen et al. compared the metabolomic profiling between primary cells isolated from 400 AML patients and those from 446 healthy controls. The study identified a metabolic signature ascribable to hypoxia, exhibiting significant alterations in six metabolites including lactate, 2-oxoglutarate, pyruvate, 2HG, and citrate, without any significant differences among different WHO AML subtypes [75]. The authors showed that the increased levels of the metabolites mentioned above, except for citrate, were found to be negatively associated with the overall survival (OS) and event-free survival (EFS) of AML patients [75].

All this evidence supports the crucial influence of metabolic switch for LSC survival in the hypoxic BMM and draws attention to the adaptive differences among the different AML subtypes. This highlights the need to further investigate and deeply understand whether the genomic background can drive the metabolic adaption of LSCs to hypoxia. Figure 3 illustrates the most relevant metabolic alterations driven by hypoxia in leukemic cells.

### 5.1. Effects of Altered Metabolism on Epigenetic Regulation

The metabolic switch induced by hypoxia is particularly relevant in light of the critical role of some TCA intermediates as epigenetic regulators. As showed in Figure 4, increased levels of 2-HG, succinate, and fumarate could be effectively used as a fuel for cancer cells to promote their growth and proliferation. Since isocitrate dehydrogenase (*IDH*) gene mutations were identified in AML, their impact on metabolism became evident. Indeed, mutant *IDH1* or *IDH2* lose their canonical function for converting isocitrate to α-ketoglutarate (α-KG), acquiring the neomorphic activity to catalyze the production of 2-HG [76]. This oncometabolite is structurally similar to α-KG; thereby, it competitively inhibits α-KG-dependent enzymes, deregulating the TCA cycle and inducing histone- and DNA-hypermethylation inhibition of epigenetic regulators [77,78]. The final consequence of 2-HG over-production in IDH-mutant AML is impaired cellular growth and differentiation arrest [79,80]. Concerning 2-HG production observed under hypoxia, it has been reported that in this condition, malate dehydrogenase, lactate dehydrogenase, and phosphoglycerate dehydrogenase could generate 2-HG, in the absence of *IDH* mutations, via enzymatic reduction of α-KG [81]. Succinate and fumarate were also classified as oncometabolite. Both were identified as regulators of some HIF targets via ten-eleven translocation (TET) inhibition [82] and as inhibitors of PHDs [83,84], suggesting a role in the maintenance of hypoxia response as well as in histone and DNA demethylation. Moreover, the succinate dehydrogenase (SDH) complex enzyme, also called mitochondrial complex II, catalyzes the oxidation of succinate to fumarate in the TCA cycle. It represents a metabolic hub for mitochondrial metabolism, also catalyzing the reduction of ubiquinone (UQ) to ubiquinol (UQH_2_), thereby linking the TCA cycle to the electron transport chain.

### 5.2. Role of Altered Metabolites in Cellular Signaling

Emerging evidence demonstrates that several metabolic intermediates could function as signaling molecules that control signals and stress response (Figure 4). For example, accumulation of the glycolytic end-product lactate under hypoxia is required for *N*-Myc downstream-regulated 3 (NDRG3) protein function. This, in turn, mediated the activation of RAF/extracellular-signal-regulated kinase 1 and 2 (ERK1/2) signaling pathways to promote angiogenesis and cell proliferation [85,86]. Additionally, increased levels of succinate and α-KG have been linked with signaling functions given their unexpected ability to act as ligands for the renin-angiotensin system through the G-protein-coupled receptors (GPR91) [87]. The activation of GPR91 mediated by succinate stimulated the proliferation of HSCs through the induction of ERK1/2 and induced multilineage blood cell recovery after chemotherapy in mice models [88]. Furthermore, the oncometabolite 2-HG also acts as a signaling metabolite by inhibiting the activity of ATP synthase. Thus, it reduces the mitochondrial respiration and mammalian target of rapamycin (mTOR) signaling [89]. These pieces of evidence promote the view that mitochondrial metabolites have a dual role in the cell: on one side, they are involved in energy metabolism; on the other side, they exhibit important signaling functions.

## 6. Gene Expression Regulation Mediated by HIFs to Sustain AML

The role of HIFs proteins as transcription factors has also been associated with their ability to regulate the expression of genes that, in turn, support leukemic transformation and progression. Thus, the communication between BMM and LSCs acts on different cellular functions to sustain AML development and progression and is recognized as a major cause of relapse.

*VEGF* represents one of the first identified targets of HIF-1α. It is a signal protein mainly involved in both physiologic and pathologic angiogenesis, and its up-regulation has been associated with several cancers, including leukemia. In particular, two different studies have confirmed that HIF-1α and HIF-2α are significantly associated with *VEGF-A* expression and BM angiogenesis, which is crucial for the pathophysiology of AML [90,91]. Moreover, it has been reported that LSCs also express the VEGFR-2 receptor in vitro and in vivo, leading to an autocrine loop that increases cellular proliferation [92].

Another well-documented HIF-1α target is represented by C-X-C motif chemokine ligand 12 (CXCL12) and its cognate receptor C-X-C motif chemokine receptor 4 (CXCR4), that are mainly involved in cellular homing of HSCs and LSCs in the BM niche [93]. It was demonstrated that *CXCR4* over-expression contributes to chemoresistance of leukemic cells [94] and predicts poor prognosis in both wild-type and FMS-like tyrosine kinase 3 (*FLT3*)-mutated AML [95,96].

Additionally, it has been demonstrated that HIF-1α induces the expression of macrophage migration factor (*MIF*) [97], with drives interleukin-8 (IL-8) production by the BM mesenchymal stromal cells (BM-MSC), thereby supporting AML cell survival and proliferation [98]. Recently, HIF-1α has been shown to directly induce the expression of IL-8 in AML cell lines and in primary AML blasts, which in turn supports survival and proliferation of AML cells [99].

Severe hypoxia, at 1% but not at 6% O_2_, down-regulated FLT3 expression in normal CD34 + hematopoietic progenitors in AML primary samples independently from the mutational state of *FLT3* [100]. Under such conditions, the survival mechanisms of LSCs were based on phosphoinositide 3-kinase (PI3K)/protein kinase B (PKB, also known as AKT) signaling that it could be activated through signal transducer and activator of transcription 5 (STAT5)-mediated up-regulation of the tyrosine kinase receptor AXL [101].

## 7. Clinical Assessment and Prognostic Value of Hypoxia

As mentioned in the above sections, the alterations induced by hypoxic BMM play a pivotal role in AML, also suggesting the potential impact of HIFs on clinical disease features. Many studies have demonstrated a prognostic role of HIF proteins and some of their targets in recent years. HIF-1α has emerged as a key regulator in tumor progression and is associated with poor prognosis in many cancers, including AML.

Immunohistochemistry analysis of primary cells isolated from 84 normal karyotype AML patients revealed that high HIF-1α levels were associated with poorer OS and EFS independently from *FLT3* internal tandem duplication (ITD) or *NPM1* mutations [66]. The higher expression levels of HIF-1α and their association with prognosis were confirmed by mRNA quantification when newly diagnosed AML were compared to healthy controls [102].

More recently, it has been reported that lactate dehydrogenase (LDH) levels could be used as a biomarker to assess the risk of transplantation in AML [103].

As mentioned above, severe hypoxia (1% O_2_) reduces the proliferative activity of primary blasts entailing a cell cycle block in the G0 phase. Thus, the quiescence state of LSCs might influence their susceptibility toward chemotherapy, which is highly dependent on the S phase for antileukemic activity. In agreement with this observation, Drolle and colleagues demonstrated that the efficacy of cytarabine arabinoside (Ara-C) was strongly reduced in AML cell lines cultured under hypoxic conditions. Moreover, they showed that hypoxic AML cells displayed higher apoptosis-inhibitory protein X-linked inhibitors of apoptosis (XIAP), together with the activation of the pro-survival PI3K pathway, demonstrating that the protective mechanism of hypoxia against Ara-C could be abrogated by PI3K inhibition [104]. Hypoxic HIF-1α stabilization decreased the in vitro sensitivity of AML cell lines to Ara-C, which was reversed by blocking O_2_ consumption and ROS production with the mitochondrial complex-III inhibitor antimycin [105,106]. Accordingly, it has been reported that AML chemo-resistant LSCs preferentially localize in the hypoxic endosteal regions of the mouse BM where they are protected from the pro-apoptotic effect of Ara-C [107].

Lastly, recent data suggest that HIF-1α could be involved in promyelocytic leukemia/retinoic acid receptor-α (PML-RARα)-dependent resistance. Indeed, in vitro and in vivo studies performed in acute promyelocytic leukemia (APL) models demonstrated that HIF-1α levels increased upon the treatment with all-trans retinoic acid (ATRA). In addition, the pharmacologic inhibition of HIF-1α using EZN-2968 displayed a synergistic effect on the impairment of APL engraftment and progression [108].

## 8. Therapeutic Opportunities: Targeting Hypoxia in AML

In light of the crucial role of hypoxia in AML development, progression and therapy response, many studies have been assessing its potential for antileukemic therapies. Two different approaches are being explored. The first one focuses on targeting downstream targets of HIFs required for survival of hypoxic LSCs; the second approach is based on hypoxia-activated prodrugs (HAPs), including some selectively activated drugs hypoxia. Therefore, in the following sections, we provide an update on these promising new therapies (Table 1).

### 8.1. Targeting HIF Downstream Genes

BL8040 is a new-generation CXCR4 inhibitor able to induce the mobilization of AML cells from the protective niche into the circulation and to promote AML differentiation and apoptosis through down-regulation of B-cell lymphoma 2 (BCL-2), myeloid cell leukemia 1 (MCL-1) and cyclin D1 as well as the AKT/ERK signaling pathway [109]. In addition, in vivo studies have shown that BL8040 synergizes either with BCL-2 or FLT3 inhibitors improving survival and reducing minimal residual disease in mice [109]. Furthermore, the safety and efficacy of BL8040 combined with high-dose cytarabine were assessed in a phase IIa study in relapsed and refractory AML, demonstrating a significantly extended OS [110] (ClinicalTrials.gov Identifier: NCT01838395). Moreover, the BATTLE study, a phase Ib/II multicenter ongoing study in AML patients aged 60 years or older, evaluates the safety, tolerability, and efficacy of BL8040 in combination with the programmed death-ligand 1 (PD-L1) inhibitor Atezolizumab (ClinicalTrials.gov Identifier: NCT03154827). Specifically, this combined approach aims to reduce the MRD status of AML patients further and prolong the period of remission.

The neutralizing monoclonal antibody (mAb) IMC-1C11 is a specific inhibitor of human VEGFR-2. Preclinical studies performed in AML models have shown that it can inhibit leukemic cell survival in vitro. Injection of IMC-1C11 in mice xenotransplanted with primary leukemic cells or AML cell lines significantly increased survival by inhibiting the proliferation of leukemic cells [92].

### 8.2. Targeting Altered Metabolism

IACS-010759 is a small-molecule inhibitor of complex I of the mitochondrial electron transport chain highly effective against OXPHOS. Preclinical studies have shown that IACS-010759 inhibits proliferation and induces apoptosis in leukemic cells, reducing tumor growth in AML mice models [111]. Based on these results, IACS-010759 was moved into phase I clinical evaluation in relapsed or refractory AML patients (ClinicalTrials.gov Identifier: NCT02882321).

Another interesting class of compounds is represented by monocarboxylate transporter (MCT) inhibitors. AR-C155858 and syrosingopine act on lactate metabolism by blocking MCT1 and MCT4 transporters, which remove the excess lactate produced by cancer cells. The lactate transporter MCT4 has been identified as a transcriptional target of HIF-1α [112] and a study from Saulle et al. demonstrated that in AML patients MCT4 over-expression is associated with poor prognosis [113]. In addition, AR-C155858 and syrosingopine exert anti-proliferative effects on AML cell lines that showed increased intracellular lactate level and enhanced the sensitivity of AML cell lines to Ara-C treatment [113].

### 8.3. Hypoxia-Activated Prodrugs (HAP)

The hypoxia-activated prodrugs approach is based on the ability of such molecules to function as O_2_ sensors so that they are activated by enzymatic reduction, specifically in hypoxic cells. Typically, a nontoxic prodrug can be activated by the enzymatic addition of one electron, which initiates the formation of DNA reactive species. This process can be inhibited by molecular oxygen. Therefore, HAPs were designed to provide targeted release of active drugs into the tumor microenvironment.

PR104 is a water-soluble phosphate ester pre-prodrug rapidly and systematically hydrolyzed to the corresponding alcohol, PR104A. PR104A, in turn, is activated in hypoxic cells resulting in hydroxylamine (PR-104H) and amine (PR-104M) metabolites, which are nitrogen mustards that act through the formation of DNA cross-links [114,115]. However, a hypoxia-independent mechanism of PR104 activation has also been reported mediated by the enzyme aldo-keto reductase 1C3 (AKR1C3), highly expressed in AML blasts [116,117]. A phase I/II study on relapsed and refractory AML reported severe gastrointestinal toxicity, myelosuppression, neutropenia, and infection [118] (ClinicalTrials.gov Identifier: NCT01037556). These observations suggest toxicity against normal hematopoiesis. Therefore, further studies will need to be performed to determine whether the combination of lower doses of PR-104 with low-dose chemotherapy could potentially maintain efficacy while reducing toxicity.

TH-302 is a 2-nitroimidazole-linked prodrug activated under hypoxia, producing the potent cytotoxin bromo-isophosphoramide mustard (Br-iPM), responsible for DNA cross-link formation. TH-302 was tested in vitro and in vivo, demonstrating hypoxia-selective activity against AML by decreasing HIF-1α expression, reducing the proliferation of leukemic cells, and enhancing double-strand breaks on DNA [119,120]. In AML xenograft models, TH-302 blocked disease progression and prolonged OS of mice [119]. Moreover, TH-302 showed synergistic antileukemic effects in *FLT3-ITD* AML models when combined with the tyrosine kinase inhibitor Sorafenib [120]. Based on these findings, a phase I study has been conducted in patients with relapsed or refractory AML (ClinicalTrials.gov Identifier: NCT01149915). The study confirmed that TH-302 treatment reduced the expression levels of HIF-1α. However, it showed a limited activity for the drug as a single agent [121].

## 9. Conclusions

Multiple studies are clarifying the role of hypoxic BMM in the pathogenesis of AML. Experimental and clinical data support the idea that the hypoxic microenvironment sustains LSC survival mainly through the activation of HIFs. Although the impact of BMM has not been well characterized in AML, it has been demonstrated that hypoxia regulates many relevant biological hallmarks of LSCs, summarized in Figure 5. The currently available literature highlights the fundamental importance of hypoxic BMM in supporting leukemic transformation and progression through the regulation of quiescence, self-renewal capacity, and cellular homing of LSCs. Moreover, the fitness of LSCs is enhanced by the metabolic shift and epigenetic modifications, and signaling alterations. Importantly, all these alterations could be crucial in resistance to therapy response, which is dependent on the biology of LSCs within the hypoxic BMM. Several mechanisms of hypoxia-mediated drug resistance have been described in AML, pointing to new attractive strategies based on targeting the alterations induced by hypoxia. Collectively, the data herein reviewed support the importance of a deeper understanding of the interactions between LSCs and the BMM as the sine-qua-non condition to design promising general strategies aimed at eradicating the ‘roots’ of AML, irrespective of the specific genetic alterations and sub-clonal complexity of this insidious form of leukemia.

## Figures and Tables

**Figure 1 ijms-22-06857-f001:**
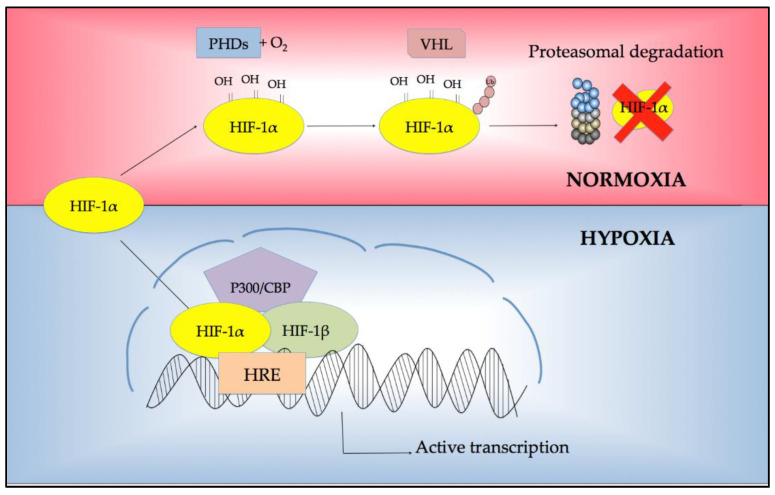
Regulation of HIF-1α protein under normoxia and hypoxia. Abbreviations: PHD: prolyl hydroxylase; VHL: Von Hippel-Lindau-associated E3-ubiquitin ligase; HIF: hypoxia-inducible factor; HRE, hypoxia-responsive element.

**Figure 2 ijms-22-06857-f002:**
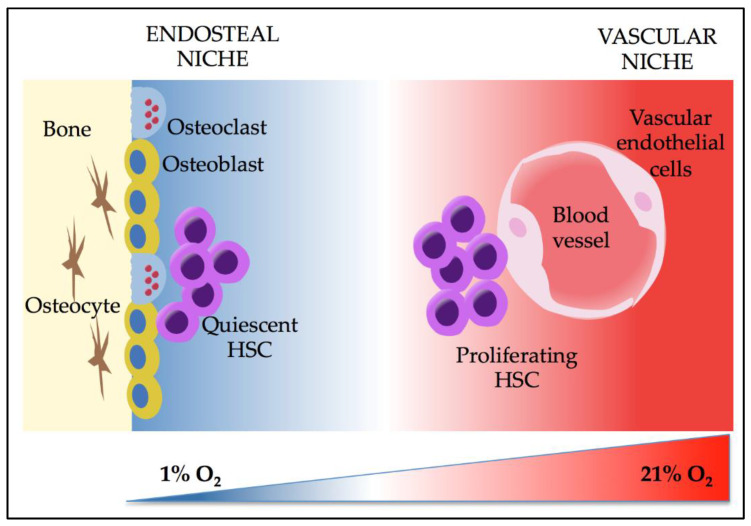
Representation of bone marrow endosteal and vascular niches with the principal cellular components and the oxygen gradient that controls HSC fate. HSC: hematopoietic stem cell.

**Figure 3 ijms-22-06857-f003:**
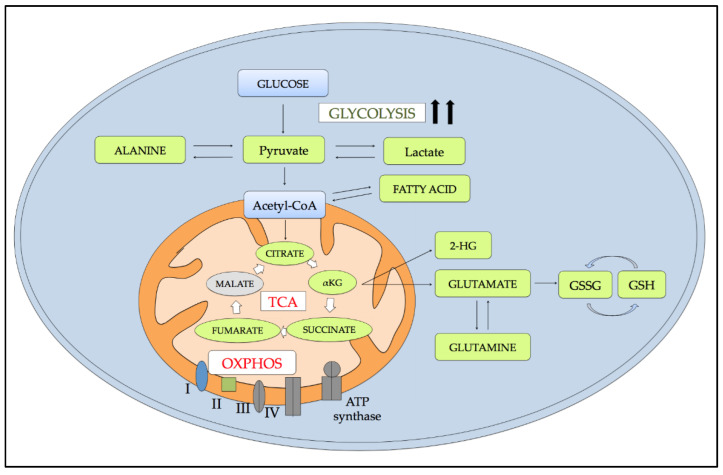
Major metabolic alterations induced by hypoxia in AML. The exposure to hypoxia induces the up-regulation (green) of glycolysis associated with increased pyruvate, lactate, and alanine levels, and the down-regulation (red) of TCA and OXPHOS responsible for TCA intermediate accumulation. Moreover, depending on cell type, hypoxic conditions can increase some detoxifying metabolites, including glutamate, reduced and oxidized glutathione (GSSG/GSH), as well as fatty acid synthesis. Abbreviations: 2-HG: 2-hydroxyglutarate; GSSG: oxidized glutathione; GSH: reduced glutathione; TCA: tricarboxylic acid; OXPHOS: oxidative phosphorylation.

**Figure 4 ijms-22-06857-f004:**
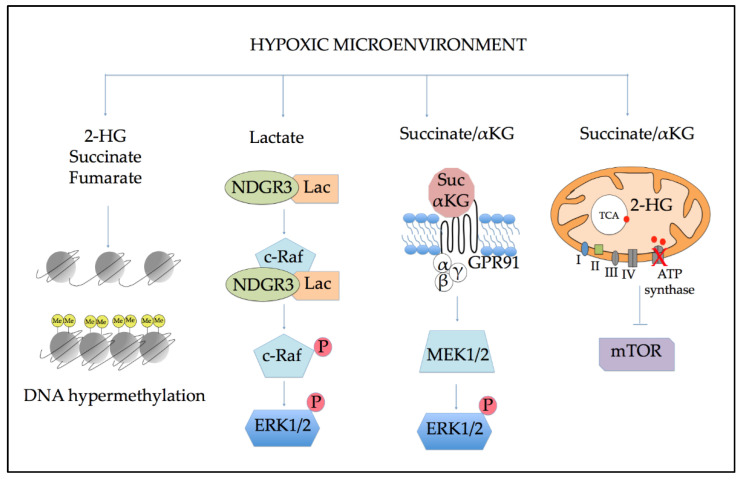
Hypoxia-altered metabolites control key cellular processes of relevance for leukemic progressions, such as epigenetic modification and signaling. Abbreviations: 2-HG: 2-hydroxyglutarate; αKG: α ketoglutarate; Lac: lactate; Suc: succinate; NDGR3: *N*-Myc downstream-regulated 3; ERK1/2: extracellular-signal-regulated kinase 1/2; GPR91: G-protein-coupled receptor 91; mTOR: mammalian target of rapamycin.

**Figure 5 ijms-22-06857-f005:**
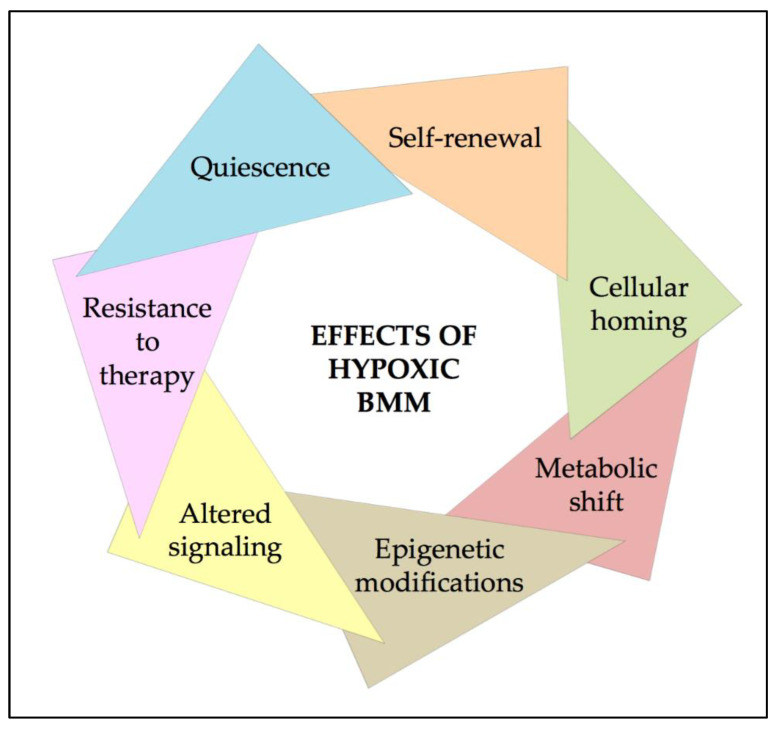
Hallmarks of hypoxia in AML.

**Table 1 ijms-22-06857-t001:** List of novel compounds developed to target LSCs in the hypoxic microenvironment. Abbreviation: na: not available.

Compound	Mechanism of Action	Clinical Trial ID	References Preclinical Studies
BL8040	CXCR4 inhibition	NCT01838395 NCT03154827	[99]
IMC-1C11	VEGFR-2 inhibition	na	[82]
IACS-010759	Complex I (OXPHOS) inhibition	NCT02882321	[101]
AR-C155858/Syrosingopine	MCT1 and MCT4 inhibition	na	[103]
PR104	DNA cross-links	NCT01037556	[104,105,108]
TH-302	DNA cross-links	NCT01149915	[109,110,111]

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
