# Peer review of "The Role of Hypoxic Bone Marrow Microenvironment in Acute Myeloid Leukemia and Future Therapeutic Opportunities"

_ijms, 2021, doi:10.3390/ijms22136857_

Round 1
Reviewer 1 Report
Having read the manuscript "The role of hypoxic bone marrow microenvironment and its therapeutic targeting in acute myeloid leukemia" I have the following comments:
- The title suggests a major emphasis of this review is the targeting of AML by therapeutic drugs, and only a small fraction of the review addresses this topic.
- Having read the review I cannot see the link between the sections that makes this a comprehensive review, it appears like 8 different sections with no common thread. Please revise to ensure there are link sentences between the sections which will make it easier for the reader to follow.
- L77 tree should be three?
- L89 what do you mean energetic metabolism?
- The authors need to extensively review the spelling and grammar in this review. Glutammine and Glutammate in Fig 3 is incorrectly spelt and there are numerous other incorrect spelling in this manuscript. I detected a mixture of British and American English in this manuscript please use one and not both forms on English throughout the manuscript.
- Your diagram of metabolism is incorrect, PDH is located in the mitochondria not cytoplasm, as acetyl CoA is produced in the mitochondrial matrix. Please correct. The two arrows showing increased glycolysis should be place to the right of the box (titled GLYCOLYSIS) not left it implies increased gluconeogenesis which is incorrect.
- P7 L267 & Fig 4. ATP synthase is not part of the Electron transport chain and should not be called Complex V. That is biochemically incorrect. In the legend of Fig 4, seuccinate is incorrectly spelt.
- L318 what is deep hypoxia? Do you not mean low hypoxia?
- L352 AKT is also known as PKB, but not serine-threonine kinase 1
- Fig 5 needs to be redrawn the text is not easily read. Also what is the bracket after the word hypoxia?
- Can you put your abbreviations in a table to help the reader.
- L407 I presume OS is overall survival?
- L425 you need to add the word "of" between "form" and "leukaemia"
- Correct reference 98
Author Response
We wish to thank the Reviewers for their constructive correction regarding our manuscript and their appreciative comments. Please see the attachment for point-by-point response.

Reviewer 2 Report
Comments on the manuscript : “The role of hypoxic bone marrow microenvironment and its therapeutic targeting in acute myeloid leukemia” by Bruno et al. (Manuscript ID: ijms-1254527)
In this review, the authors give an overview of the current literature concerning the crosstalk between leukemic stem cells and a hypoxic bone marrow microenvironment.
This version of the review is well written, clear and informative.
My main concern is that the authors need to add more recent publications in the introduction and in the two paragraphs concerning the effects of hypoxia on HSCs and Hifs and AML, in order to have a better update on the subject.
Please find below some suggestions to improve the ms
-Need to define the terms “Os” and “EFS” ( ine 311)
-A table summarizing drug characteristics, presented in lines 339-412, would be helpful.
-line 196 : need to define what cell line types are “THP1”and “NB4”
-line 242 : the authors could refer to figure 4
There are several typos:
-Fig 1 : P300 should be written on the same line
-Fig 5: there are too many errors (including signs such as * and +)
-line 77 : this is “three” instead of “tree”
Author Response
We wish to thank the Reviewers for their constructive correction regarding our manuscript and their appreciative comments. Please see the attachment for point by point response.

Round 2
Reviewer 1 Report
The revised manuscript has addressed the issues raised in my earlier review. I thank the authors for making these changes.